# Extended Texture Analysis of Non-Enhanced Whole-Body MRI Image Data for Response Assessment in Multiple Myeloma Patients Undergoing Systemic Therapy

**DOI:** 10.3390/cancers12030761

**Published:** 2020-03-24

**Authors:** Kaspar Ekert, Clemens Hinterleitner, Karolin Baumgartner, Jan Fritz, Marius Horger

**Affiliations:** 1Department of Radiology, Diagnostic and Interventional Radiology, Eberhard Karls University, Hoppe-Seyler-Str. 3, 72076 Tübingen, Germany; 2Department of Internal Medicine II, Eberhard-Karls-University, Otfried-Müller-Str. 10, 72076 Tübingen, Germany; 3Department of Radiology, Grossman School of Medicine, New York University Langone Health, 660 1st Ave, 3rd Floor, Rm 313, New York, NY 10016, USA

**Keywords:** radiomics, multiple myeloma, MRI, diffusion imaging

## Abstract

Identifying MRI-based radiomics features capable to assess response to systemic treatment in multiple myeloma (MM) patients. Retrospective analysis of whole-body MR-image data in 67 consecutive stage III MM patients (40 men; mean age, 60.4 years). Bone marrow involvement was evaluated using a standardized MR-imaging protocol consisting of T1w-, short-tau inversion recovery- (STIR-) and diffusion-weighted-imaging (DWI) sequences. Ninety-two radiomics features were evaluated, both in focally and diffusely involved bone marrow. Volumes of interest (VOI) were used. Response to treatment was classified according to International Myeloma Working Group (IMWG) criteria in complete response (CR), very-good and/or partial response (VGPR + PR), and non-response (stable disease (SD) and progressive disease (PD)). According to the IMWG-criteria, response categories were CR (*n* = 35), VGPR + PR (*n* = 19), and non-responders (*n* = 13). On apparent diffusion coefficient (ADC)-maps, gray-level small size matrix small area emphasis (Gray Level Size Zone (GLSZM) small area emphasis (SAE)) significantly correlated with CR (*p* < 0.001), whereas GLSZM non-uniformity normalized (NUN) significantly (*p* < 0.008) with VGPR/PR in focal medullary lesions (FL), whereas in diffuse involvement, 1st order root mean squared significantly (*p* < 0.001) correlated with CR, whereas for VGPR/PR Log (gray-level run-length matrix (GLRLM) Short Run High Gray Level Emphasis) proved significant (*p* < 0.003). On T1w, GLRLM NUN significantly (*p* < 0.002) correlated with CR in FL, whereas gray-level co-occurrence matric (GLCM) informational measure of correlation (Imc1) significantly (*p* < 0.04) correlated with VGPR/PR. For diffuse myeloma involvement, neighboring gray-tone difference matrix (NGTDM) contrast and 1st order skewness were significantly associated with CR and VGPR/PR (*p* < 0.001 for both). On STIR-images, CR correlated with gray-level co-occurrence matrix (GLCM) Informational Measure of Correlation (IMC) 1 (*p* < 0.001) in FL and 1st order mean absolute deviation in diffusely involved bone marrow (*p* < 0.001). VGPR/PR correlated at best in FL with GSZLM size zone NUN (*p* < 0.019) and in all other involved medullary areas with GLSZM large area low gray level emphasis (*p* < 0.001). GLSZM large area low gray level emphasis also significantly correlated with the degree of bone marrow infiltration assessed histologically (*p* = 0.006). GLCM IMC 1 proved significant throughout T1w/STIR sequences, whereas GLSZM NUN in STIR and ADC. MRI-based texture features proved significant to assess clinical and hematological response (CR, VPGR, and PR) in multiple myeloma patients undergoing systemic treatment.

## 1. Introduction

Multiple myeloma is a malignant hematologic disease of the mature B-cells with a still unfavorable prognosis despite significant advances in treatment [1,2]. It primarily involves the bone marrow where an increasing tumor burden with tumor-associated osteoclast activation and osteoblasts inhibition results in bone destruction called “myeloma bone disease” putting the patients at risk for fracture [3,4]. Bone marrow involvement takes different appearances ranging from diffuse to focal infiltration or a blend of both [5]. The degree of myeloma cell infiltration varies considerably particularly in the diffuse form impacting signal intensity on various MRI sequences [6]. By comparison to other malignant diseases, multiple myeloma can be generally well monitored by quantifying paraproteins (M-protein) in serum and urine. However, some myelomas are non-secretory or hypo-secretory and are therefore difficult to manage both in the primary diagnosis, as well as during therapy [7]. Imaging of multiple myeloma was initially applied for classification according to the Durie & Salmon criteria published in 1975 focusing on bone destruction as a surrogate for the myeloma cell burden in addition to diagnosing complications, such as fractures or detecting potential causes for neurologic symptoms [8,9]. With the advent of whole-body MRI, imaging concentrated increasingly on the assessment of medullary and extra-medullary involvement [10]. Whereas, primarily detection and surveillance of focal-nodular medullary lesions seems to be more easily done, in case of diffuse bone marrow involvement, signal intensity changes vary considerably and overlap with those of normal bone marrow patient thus illustrating inter-individual variability, which is primarily influenced by the patient’s age [11]. Both qualitative MRI-monitoring using ancillary sequences like T1-weighted (T1w) and T2w and quantitative monitoring using functional image data derived from diffusion-weighted-imaging (DWI) have been therefore, recommended [12]. However, their interpretation is challenging as signal intensity changes are subject to temporal variability and are dependent also on the amount of red and yellow marrow and treatment-related shifts between these two components [11]. An alternative to the classical MRI approach for monitoring myeloma response to treatment could be represented by texture analysis applied throughout all used sequences. This technique has gained increased attention in the last years being used as a complementary quantitative tool for tumor characterization and evaluation of response [13,14,15].

The intention of this study was to assess the potential role of texture analysis applied on MRI-image data for response evaluation by comparison with serologic (M-protein) data.

## 2. Material and Methods

### 2.1. Patient Characteristics

This was a retrospective study which was approved by our institutional review board and registered under the number 302/2019BO2. Patients were identified by a patient chart search at our institution between January 2014 and December 2018. The following patients were eligible: adults with histologically and hematologically proven stage III multiple myeloma (according to the Durie & Salmon criteria) requiring treatment, as well as having been staged by whole-body MRI [9]. The exclusion criteria were patients with other malignancies or hematologic disorders; contraindications to MRI, such as claustrophobia, implanted pacemakers and patients examined by using another MR imaging protocol. Sixty-seven consecutive patients (40 men and 27 women; median age, 60.4 years; age range, 42.3–78.6 years) were evaluated which meet the inclusion criteria of baseline whole-body MRI and standard hematologic monitoring in a 3 months cycle. No patient received radiation therapy at or adjacent to the skeletal site that was monitored by MRI. Patients were diagnosed with IgG κ (*n* = 31), IgA κ (*n* = 8), IgG λ (*n* = 10), IgA λ (*n* = 11), light-chain myeloma (*n* = 7), and non-secretory myeloma (*n* = 0).

For primary therapy, 19 patients received Bortezomib/Cyclophosphamide/Dexamethasone, 25 patients Bortezomib/Leneladomide/Dexamethasone, 9 patients Bortezomib/Lineladomide/Dexamethasone + Elotuzumab, 8 patients Bortezomib/Doxorubicin/Dexamethasone, and 6 patients Isatuximab/Lenalidomide/Carfilzomib/Dexamethasone.

Forty-nine of 67 patients underwent autologous stem cell transplantation.

Mean follow-up interval was 11.6 months (range, 3.4–25.1 months).

### 2.2. Whole-Body DWI Protocol Design

Examinations have been performed on a 1.5 T MR scanner (Magnetom Avanto, Siemens Healthineers) both at baseline (before treatment onset) and after treatment using a minimum of three sequences (T1w-, STIR-, and DWI) (details are available in Appendix A).

### 2.3. Image Analysis

Focal lesions, as well as diffuse myeloma manifestations, in the skeleton were segmented by volumes of interest (VOI) (Figure 1 and Figure 2). A standardized spherical VOI of 3.8 mL was placed in the wing of ilium to correlate bone marrow biopsy and thus response criteria to diffuse bone marrow texture analysis.

Texture feature analysis was performed using the pyradiomics library. Only original order features were included, filtered features (i.e., wavelets, square root) were not included due to the small sample size. Altogether, 92 features were extracted and used for texture analysis (refer to Appendix A).

### 2.4. Standard of Reference

Response categories were defined according to the International Myeloma Working Group (IMWG) based on the M-protein level in serum and urine indicating paraproteinemia and paraproteinuria. Responders were classified into complete response (35/54) and near complete response and very good partial response (13/54) + partial response (6/54). Non-responders were composed of stable (3/13) and progressive disease (10/13). Therapy response was not evenly distributed with 54 responders and 13 non-responders.

In a subgroup of patients (33/67) and at 55/107 time points, radiomics parameters were also correlated with the degree of myeloma cell infiltration of the bone marrow.

### 2.5. Laboratory Data

For each patient, the levels of serum and urine M-protein were determined at the time of diagnosis and at follow-up during and after anti-myeloma treatment. The latter was used for assessment of disease response including immunofixation for confirmation of complete tumor response. At our institution, the normal values for hematologic parameters determined by the laboratory are IgG, 700–1600 mg/dL; IgA, 70–400 mg/dL; IgM, 40–230 mg/dL; serum light chains λ, 8.1–33.0 mg/L; and light chains κ, 3.6–15.9.

### 2.6. Statistical Analysis

Continuous variables are presented as median and 95% CI, categorical variables are given by number and percentages. Binary logistic regression analysis was used in all cases (*n* = 67) to identify variables significantly associated with complete response (CR), VGPR (very good partial response), or partial response (PR), defined according to the IMWG guidelines. Prior regression analysis log transformation was performed in all structural parameters. In the case of negative values, the following adjustment equation was used: log_adj._(y.) = log(y) − log(ymin) + 1

In order to perform a trend analysis, baseline variables were established using the first MRI measurement in each subgroup. All significant variables were analyzed in a second step using Wilcoxon matched-pairs signed rank test. Finally, the predictive values of all identified parameters were evaluated by examining the area under the receiver-operator characteristic (ROC) curve using a confidence interval of 95%. All tests were considered statistically significant when *p* < 0.05. Statistical analyses were computed using SigmaStat, version 21 (SPSS).

## 3. Results

### 3.1. Identification of MRI-Features (ADC) Associated with Hematological Outcomes (CR, PR, or VGPR)

In order to identify MRI textural features significantly correlated with complete response (CR) and response to treatment (PR, VGPR), binary logistic regression was used for all 92 textural features. Significant variables were analyzed then, using a Wilcoxon matched-pairs signed rank test, in a second step. In patients achieving complete remission (CR), Log (Gray Level Size Zone Matrix (GLSZM) Small Area Emphasis) quantified in areas of focal MM involvement correlated significantly (baseline 0.1, 95% CI: 0–0.14 vs. CR 0.14, 95% CI: 0.09–0.21; *p* < 0.001). In areas with diffuse bone marrow infiltration, Log (1st order Root Mean Squared) significantly correlated with CR (baseline 2.7, 95% CI: 2.25–3.01 vs. CR 2.51, 95% CI: 2.1–2.89; *p* < 0.001).

In focal lesions of patients achieving PR or VGPR, Log_adj._ (GLSZM Size Zone Non Uniformity Normalized) was found significantly elevated compared to baseline (baseline 1.1, 95% CI: 0.63–1.26 vs. PR/VGPR 1.22, 95% CI: 1–1.55; *p* = 0.0089), whereas in areas of diffuse involvement, Log (GLRLM Short RunHigh Gray Level Emphasis) was shown to significantly decrease over baseline in patients reaching PR or VGPR during treatment (baseline 2.54, 95% CI: 1.75–3.24 vs. PR/VGPR 2.27, 95% CI: 1.33–2.64; *p* = 0.003). The correlative values of all identified texture variables were evaluated by examining the area under the receiver-operator characteristic (ROC) curve (Figure 3A–D).

### 3.2. Identification of MRI-Features (T1w) Associated with Hematological Response Categories (CR, PR, or VGPR)

MRI textural features which were significantly associated with a complete remission (CR) and response (PR, VGPR) on T1w-images were likewise identified using binary logistic regression analysis. In patients achieving complete response (CR), Log_adj._ (GLRLM Run Length Non Uniformity Normalized) was significantly elevated in focal lesions (baseline 0.1, 95% CI: 0.01–0.15 vs. CR 0.12, 95% CI: 0.046–0.177; *p* = 0.03), whereas in areas of diffuse bone marrow infiltration, Log (Ngtdm Contrast) was shown to be significantly associated with CR (baseline 0.43, 95% CI: 0.27–0.52 vs. CR 0.43 95% CI: 0.28–0.7; *p* < 0.001).

In focal lesions, Log_adj._ (Glcm Imc1) was significantly elevated compared to baseline (baseline 0.1, 95% CI: 0.01–0.15 vs. PR/VGPR 0.122, 95% CI: 0.05–0.177; *p* = 0.048) in patients achieving PR or VGPR Whereas in areas of diffuse bone marrow involvement, Log (1st order Skewness) was shown to significantly decrease in patients responding to treatment (baseline 0.46, 95% CI: 0.28–0.67 vs. PR/VGPR 0.41, 95% CI: 0.13–0.53; *p* < 0.001). The predictive values of all identified texture variables were evaluated by examining the area under the receiver-operator characteristic (ROC) curve (Figure 4A–D).

### 3.3. Identification of MRI-Features (T2w) Associated with Clinical Outcomes (CR, PR, or VGPR)

MRI textural features which were significantly associated with complete response (CR) and response (PR, VGPR) on T2w-images are shown in Figure 5. In patients with CR, Log_adj._ (Glcm Imc1) was significantly elevated in focal lesions (baseline 0.09, 95% CI: 0.04–0.16 vs. CR 0.13, 95% CI: 0.08–0.198; *p* < 0.001) compared to baseline, whereas in areas with diffuse bone marrow involvement, Log (1st order Mean Absolute Deviation) was shown to be significantly associated with CR (baseline 1.1, 95% CI: 0.41–1.44 vs. CR 0.66 95% CI: 0.24–1.18; *p* < 0.001). In patients achieving PR or VGPR Log_adj_. (GLSZM Size Zone Non Uniformity Normalized) proved significantly elevated compared to baseline (baseline 0.12, 95% CI: 0.001–0.25 vs. PR/VGPR 0.17, 95% CI: 0.08–0.34; *p* = 0.014) in focal lesions, whereas in areas with diffuse bone marrow involvement, Log (GLSZM Large Area Low Gray Level Emphasis) showed a significantly decrease in patients reaching PR or VGPR (baseline 4.9, 95% CI: 3.67–6.26 vs. PR/VGPR 5.75, 95% CI: 4.55–7.1; *p* < 0.001) after treatment. The ROC analyses are given in Figure 4A–D.

### 3.4. Correlation of MRI-Features with Biopsy Confirmed Degree of Myeloma Cell Infiltration

GLSZM large area low gray level emphasis in T1w sequences measured in the wing of ilium showed a significant correlation (*p* = 0.006; *r* = 0.19) with the degree of bone marrow infiltration by myeloma cells confirmed by bone marrow biopsy (Figure 6). Bone marrow infiltration confirmed by biopsy showed a mean infiltration of 32.1% (range, 1–100%; median 25%).

## 4. Discussion

Our results indicate that ten radiomics features quantified on T1w-, ADC- and STIR- images may be associated with the depth of myeloma response to systemic therapy as classified by the IMWG response criteria [16]. Statistically significant textural features varied throughout the applied MR-sequences in a similar fashion as observed with the visual assessment of ancillary imaging findings in myeloma patients. Hence, on T1w images, the gray-level run-length matrix (GLRLM) non-uniformity normalized-NUN significantly increased in CR in focal lesions, whereas the neighboring gray-tone difference matrix (NGTDM) contrast turned out to be significantly associated with CR in areas with diffuse medullary involvement, where it also significantly increased over baseline. Both features reflect lower tissue homogeneity following treatment compared to the mean values of the corresponding baseline cohort. These results presumably reflect lower levels of myeloma cell infiltration with interspersed yellow marrow accounting for larger ranges of gray values and less similarities of run lengths throughout the image. In patients experiencing partial or very good partial response (exemplary case Figure 1 and Figure 2), the levels of gray-level co-occurrence matric (GLCM) informational measure of correlation (Imc1) increased, whereas the 1st order skewness decreased after treatment. These two features reflect higher levels of tissue homogeneity in different ways, presumably consistent with higher degrees of myeloma cell infiltration before and also after treatment compared to the baseline cohort. Again, these findings may present a plausible explanation for the limited response in this subgroup. On ADC maps, an increase in gray-level small size matrix small area emphasis (GLSZM SAE) was significantly associated with CR in focal lesions, whereas a decrease in 1st order root mean squared was found in areas of diffuse medullary involvement. Interestingly, these two features reflect lower ADC-values and a coarser image texture at follow-up, which could also be explained, e.g., by lower myeloma cellularity and lower tissue homogeneity due to interspersed fatty marrow, which knowingly exhibits even lower ADC-values compared to the tumor. As a great overlap is expected between the ADC-values in involved bone marrow and non-involved fatty marrow (both exhibiting low water proton diffusivity), the features found significantly associated with PR/VGPR both reflect lesser tissue homogeneity. Hence, GLSZM non-uniformity normalized (NUN) significantly correlated with VGPR/PR in focal medullary lesions, whereas in diffuse involvement gray-level run-length matrix (GLRLM) non-uniformity normalized (NUN) proved significant. Interestingly, this trend changed on STIR sequences with an increase in gray-level co-occurrence matrix (GLCM) Informational Measure of Correlation (IMC) 1 and a decrease of 1st order mean absolute deviation proving significantly associated with complete response in both focal and diffuse marrow involvement. These features indirectly express a higher degree of tissue homogeneity, which may be interpreted as a consequence of fat saturation, which is inherent to the STIR sequence, thus minimizing the contrast between neighboring voxels which is otherwise expected in case of intermixed fatty marrow.

Of note, GLSZM large area low gray level emphasis significantly correlated with the degree of bone marrow infiltration assessed histologically by bone marrow biopsy (Figure 6). Exemplarily highlighted in the treatment course of two patients (Figure 7) alternating between progression and response, which is also reflected by our textural analysis.

The temporal course of GLSZM large area low gray level emphasis in T1w sequences measured in the wing of ilium is demonstrated parallel to the response categories in two exemplarily patients.

Imaging markers may provide additional prognostic value in myeloma patients undergoing systemic treatment. By now, MRI has evolved as the favored imaging technique with respect to visualization and characterization of bone marrow changes [17,18]. For a long time, bone marrow MR-imaging was based on visual (qualitative) assessment of signal intensity levels on T1w- and T2-w scans alone [19]. However, the great inter-individual signal variability impedes this approach particularly in lower degrees of marrow infiltration. Moreover, a great overlap between normal and abnormal bone marrow signal was reported, limiting MRI-specificity [19]. Notably, age and treatment have considerable impact on the medullary appearance generating equivocal findings which could not be entirely differentiated up to now [11]. Diffuse medullary involvement accounts for one of the most challenging aspects in myeloma monitoring due to lower cell counts compared to the nodular focal lesions [20]. Moulopoulos et al. found diffuse marrow involvement was more frequently associated with high-risk cytogenetics and a worse prognosis [20].

Quantification of water proton diffusivity (DWI) on corresponding ADC-maps has been praised as an alternative to qualitative visual assessment and promising results have been published on this topic but mainly addressing short-term monitoring and focal lesions [21,22]. However, when DWI is applied, great overlap in ADC-values were found both in baseline setting in lower degrees of medullary involvement, as well as while undergoing therapy, where with receding tumor infiltration ADC values continued to strongly decrease due to emerging fatty marrow [23].

A potential increase in accuracy of MR image interpretation could be facilitated by radiomics features which may quantify structural characteristics of bone marrow changes. Giles et al. demonstrated that whole-body DWI was reliable for treatment response evaluation in myeloma patients [24]. The histogram metrics of 1st order features quantified in this study showed significant differences identifying CR in diffuse involvement in myeloma patients. Other reports proposed prognostic factors related to changes induced by systemic treatment on ADC-values early or at mid-term [22,23]. Histogram-based prediction of response in myeloma patients has also been applied on ^18^F-Fluordesoxyglucose (^18^F-FDG) Positron emission tomography (FDG-PET) image data using machine learning, indicating quantitative heterogeneity may reduce the error of the predicted progression [25].

Our study has some limitations. First, the cohort was retrospectively evaluated and composed of heterogeneous myeloma subtypes, as well as magnitude of M-protein. Second, patients underwent different treatment regimens which are not comparable. Third, there were only few non-responders in our cohort evaluating prognosis based on radiomics data in this subgroup non-assessable. Fourth, due to the small number of patients in this series and the fact that all subjects were examined on only one scanner, technical data reproducibility should be confirmed in larger patient multi-center studies.

## 5. Conclusions

In conclusion, MRI-based textural features proved to correlate well with the clinical and hematological response (CR, VPGR, and PR) in multiple myeloma undergoing systemic treatment and may therefore be implemented as a complementary prognosis evaluation tool in assessing myeloma patients’ prognosis. One textural feature (GLSZM large area low gray level emphasis) correlated also with the degree of bone marrow infiltration confirmed by biopsy.

## Figures and Tables

**Figure 1 cancers-12-00761-f001:**
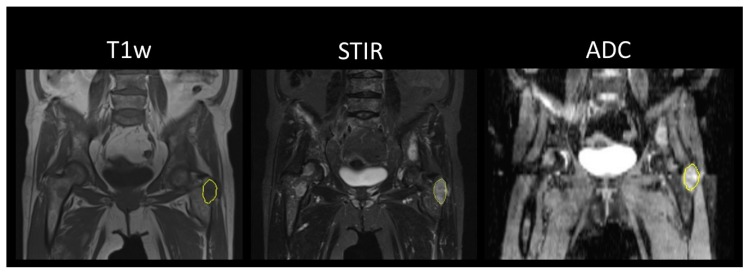
Exemplary case in a 66-year-old female myeloma patient prior to treatment, stage IIIa (according to the Durie & Salmon criteria). The focal myeloma lesion in the left greater trochanter is segmented on T1w and STIR sequence, as well as on apparent diffusion coefficient (ADC) maps.

**Figure 2 cancers-12-00761-f002:**
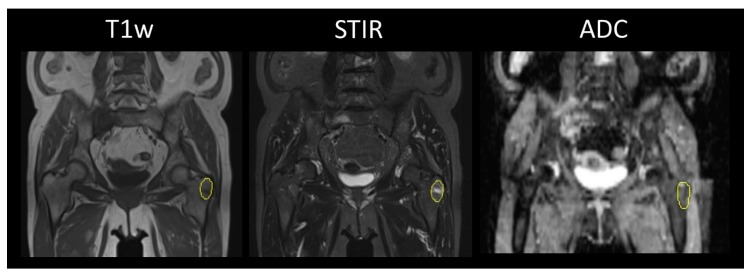
Sixty-six-year-old female patient from Figure 1 demonstrating complete response to systemic therapy with bortezomib, lenalidomide, and dexamethasone according to the International Myeloma Working Group (IMWG). However, the focal lesion in the left greater trochanter is still showing a heterogenous signaling pattern in all three sequences.

**Figure 3 cancers-12-00761-f003:**
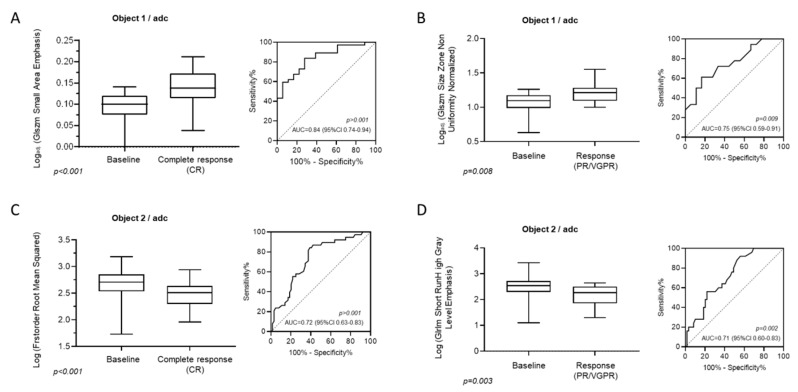
Box plots on the left presenting the statistically significant textural features on ADC images both in focal (**A**,**B**) and diffuse marrow involvement (**C**,**D**) both for patients experiencing complete response (**A**,**C**) and very good partial response/partial response(**B**,**D**). The receiver-operator characteristic (ROC) analysis on the right illustrates the area under the curve (AUC), corresponding *p*-value and confidence interval for the log of the analyzed texture feature. GLRLM = gray-level run-length matrix.

**Figure 4 cancers-12-00761-f004:**
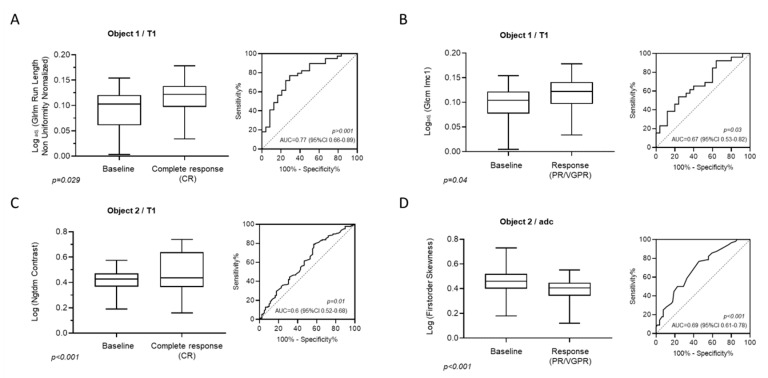
Box plots presenting the statistically significant textural features on T1-weighted images both in focal (**A**,**B**) and diffuse marrow involvement (**C**,**D**) both for patients experiencing complete response (**A**,**C**) and very good partial response/partial response (**B**,**D**). The ROC analysis on the right illustrates the AUC, corresponding *p*-value and confidence interval for the log of the analyzed texture feature.

**Figure 5 cancers-12-00761-f005:**
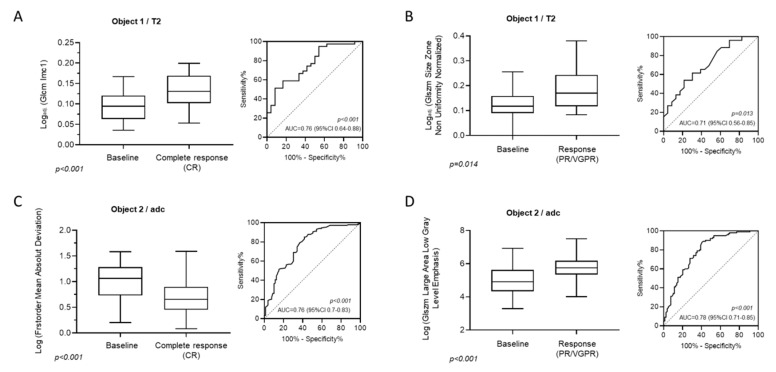
Box plots presenting the statistically significant textural features on STIR images both in focal (**A**,**B**) and diffuse marrow involvement (**C**,**D**) both for patients experiencing complete response (**A**,**C**) and very good partial response/partial response (**B**,**C**). The ROC analysis on the right illustrates the AUC, corresponding *p*-value and confidence interval for the log of the analyzed texture feature.

**Figure 6 cancers-12-00761-f006:**
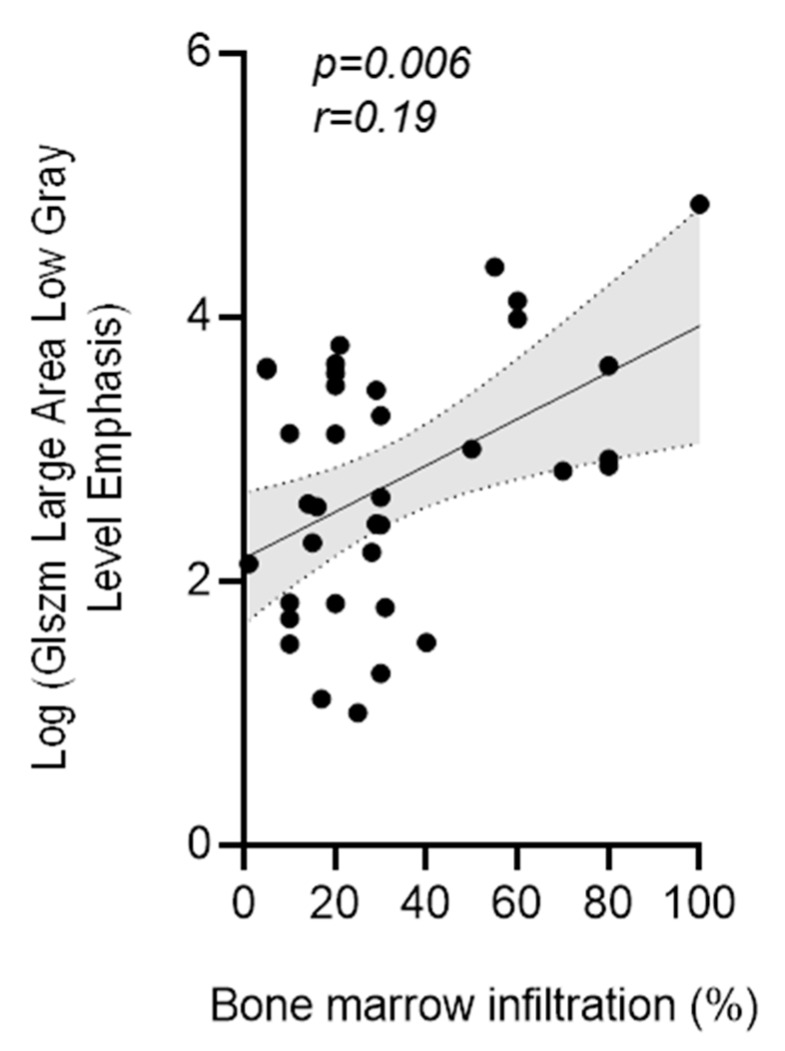
Correlation between the course of GLSZM large area low gray level emphasis and M-protein with the degree of bone marrow infiltration GLSZM large area low gray level emphasis in T1w sequences measured in the wing of ilium showed significant (*p* = 0.006; *r* = 0.19) correlation with the degree (%) of bone marrow infiltration by myeloma cells confirmed by bone marrow biopsy.

**Figure 7 cancers-12-00761-f007:**
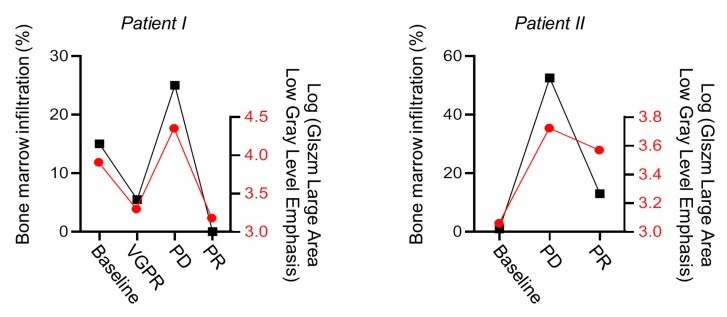
Longitudinal course of GLSZM large area low gray level emphasis and correlation with M-protein based response categories to anti-myeloma treatment. VGPR = very good partial response; PD = progressive disease; PR = partial response.

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
