# Peer review of "Extended Texture Analysis of Non-Enhanced Whole-Body MRI Image Data for Response Assessment in Multiple Myeloma Patients Undergoing Systemic Therapy"

_cancers, 2020, doi:10.3390/cancers12030761_

Round 1
Reviewer 1 Report
The authors described clinical utility of texture analysis on MRI image data for response evaluation for multiple myeloma. This is an interesting attempt but novelty and advantage of this analysis should be clearly demonstrated because similar studies have been published.
Overall, Abstract and Discussion are too long, so the authors should shorten these parts concisely and make it clear.
The authors repeatedly stated that texture features of MRI imaging are predictive for the depth of myeloma response, but depth of response is defined by M-protein in either serum or urine and proportion of plasma cell in bone marrow. The term predictive reminds readers that texture feature of MRI before treatment have predictive value for the treatment outcome, but such results are not presented. Therefore, I think "predictive and predict" should be changed to other appropriate term.
In the material and methods, why the authors analyse only multiple myeloma stage 3 by Durie and Salmon criteria? Almost all symptomatic myeloma patients according to IMWG criteria are usually treated.
In the result section, what baseline is not clearly presented. Does "Baseline" mean values collected just before treatment or other?
There are mistakes.
For examples, CR is not complete remission in multiple myeloma. CR is complete response, VGPR and PR are the same according to the uniform response criteria by IMWG.
In introduction, Simon and Durie criteria should be changed to Durie and Salmon criteria. In material and methods, Salmon and Durie shoudld be Durie and Salmon.
The term M-gradient is uncommon, should be M-protein or M-component.
Author Response
The authors described clinical utility of texture analysis on MRI image data for response evaluation for multiple myeloma. This is an interesting attempt but novelty and advantage of this analysis should be clearly demonstrated because similar studies have been published.
Overall, Abstract and Discussion are too long, so the authors should shorten these parts concisely and make it clear.
We shortened the abstract as well as discussion as much as possible.
The authors repeatedly stated that texture features of MRI imaging are predictive for the depth of myeloma response, but depth of response is defined by M-protein in either serum or urine and proportion of plasma cell in bone marrow. The term predictive reminds readers that texture feature of MRI before treatment have predictive value for the treatment outcome, but such results are not presented. Therefore, I think "predictive and predict" should be changed to other appropriate term.
We removed the noun predictive using either “associated” or “correlated”.
In the manuscript title, we changed predicting to “monitoring” response to treatment.
In the material and methods, why the authors analyze only multiple myeloma stage 3 by Durie and Salmon criteria? Almost all symptomatic myeloma patients according to IMWG criteria are usually treated.
This is correct. However, in our cohort all patients were classified stage III either based on the Durie & Salmon classification or based on hematologic parameters.
In the result section, what baseline is not clearly presented. Does "Baseline" mean values collected just before treatment or other?
Baseline means values collected before therapy onset. We added this information in the M&M section.
There are mistakes.
For examples, CR is not complete remission in multiple myeloma. CR is complete response, VGPR and PR are the same according to the uniform response criteria by IMWG.
Please excuse, we changed all response categories accordingly both in the text body and the figures.
In introduction, Simon and Durie criteria should be changed to Durie and Salmon criteria.
In material and methods, Salmon and Durie shoudld be Durie and Salmon.
As requested, we interchanged Durie & Salmon.
The term M-gradient is uncommon, should be M-protein or M-component.
We corrected M-gradient accordingly to M-protein.
Reviewer 2 Report
Title: Extended texture analysis of non-enhanced whole-body MRI image data for prediction of response in multiple myeloma patients undergoing systemic therapy
Availability of a reliable pretreatment prognostic indicator would be such a blessing in oncology.
The authors attempted to identify such imaging markers for multiple myeloma patients based on the texture features observed on MRI.
The work is rather simple and the results are somewhat complicated but would provide a foundation to move forward to the next level of investigation.
While looking forward to the reproducibility study hopefully in a larger number of patients, the MS requires minor clarification.
This was a retrospective study. The patients with stage III multiple myeloma, who required treatment and had a whole body MRI for staging were included. Contraindications to have an MRI were included in the exclusion criteria.
The tumor textures were extracted from T1WI, ADC map and STIR images using pyramidiomics library. A total of 92 features among the original order features were extracted from focal and diffuse lesions. The extracted features were correlated with clinical outcome to evaluate their prognostic values. The clinical outcome were categorized into CR, PR, VGPR-PR, Non-responder (SD+PD) based on the criteria defined by the international myeloma working group.
Q:
Methods:
- It is not clear if the patients had the whole body MRIs prior to or after accrual (this seems what happened, as an inclusion criteria).
If the former was true, the “MRI contraindications” was not needed in the exclusion criteria? It was stated as a retrospective study. Please clarify.
- T1WI, ADC map and STIR images were used for texture analysis.
It is not clear how many time points that the MRIs were acquired (e.g. pre and 3 month post treatment). The results appear compared with the “baseline”. It is not clear if the variables were identified from the pretreatment MRI or at the time of response evaluation. Please clarify.
Results:
All results were well lined up throughout the MS. It seems, though, a discrepancy is noted in one category (unless these are equivalent). Please clarify.
In the abstract,
On ADC map, in diffuse involvement, in prediction of VGPR/PR, gray-level run-length matrix (GLRLM) non-uniformity normalized (NUN) proved significant (p< 0.002).
In the results and fig 3:
On ADC map, in diffuse involvement, in prediction of VGPR/PR, gray-level run-length matrix (GLRLM) short run high gray level emphasis was significantly decreased over the baseline.
Comments: the work would be more valuable if any reproducibility data are available.
- Technical reproducibility within and between the scanners, venders
- Reproducibility in a larger group of patients
- Reproducibility between the types of myeloma
Author Response
Availability of a reliable pretreatment prognostic indicator would be such a blessing in oncology.
The authors attempted to identify such imaging markers for multiple myeloma patients based on the texture features observed on MRI.
The work is rather simple and the results are somewhat complicated but would provide a foundation to move forward to the next level of investigation.
While looking forward to the reproducibility study hopefully in a larger number of patients, the MS requires minor clarification.
This was a retrospective study. The patients with stage III multiple myeloma, who required treatment and had a whole body MRI for staging were included. Contraindications to have an MRI were included in the exclusion criteria.
The tumor textures were extracted from T1WI, ADC map and STIR images using pyramidiomics library. A total of 92 features among the original order features were extracted from focal and diffuse lesions. The extracted features were correlated with clinical outcome to evaluate their prognostic values. The clinical outcome were categorized into CR, PR, VGPR-PR, Non-responder (SD+PD) based on the criteria defined by the international myeloma working group.
Q:
Methods:
- It is not clear if the patients had the whole body MRIs prior to or after accrual (this seems what happened, as an inclusion criteria).
Whole-body MRI was performed both at baseline and after treatment. Radiomics analysis was applied to both image data at both time points highlighting the significant changes in the feature magnitude. We added this information in the M&M section.
If the former was true, the “MRI contraindications” was not needed in the exclusion criteria? It was stated as a retrospective study. Please clarify.
This was indeed a retrospective data evaluation using radiomics features. Nonetheless, imaging monitoring of these patients was part of the routine diagnosis and all mentioned exclusion criteria refer to standard contraindications for MRI.
- T1WI, ADC map and STIR images were used for texture analysis.
It is not clear how many time points that the MRIs were acquired (e.g. pre and 3 month post treatment). The results appear compared with the “baseline”. It is not clear if the variables were identified from the pretreatment MRI or at the time of response evaluation. Please clarify.
We evaluated radiomics features both at baseline and after treatment and calculated all significant changes between the two time points using the hematological response as a ground truth. Hence, all statistically significant changes in the magnitude of radiomics features between baseline and post-treatment were correlated with the hematological response categories (2.6. Statistical analysis).
Results:
All results were well lined up throughout the MS. It seems, though, a discrepancy is noted in one category (unless these are equivalent). Please clarify.
In the abstract,
On ADC map, in diffuse involvement, in prediction of VGPR/PR, gray-level run-length matrix (GLRLM) non-uniformity normalized (NUN) proved significant (p< 0.002).
In the results and fig 3:
On ADC map, in diffuse involvement, in prediction of VGPR/PR, gray-level run-length matrix (GLRLM) short run high gray level emphasis was significantly decreased over the baseline.
Many thanks for this correction! It was a data transmission failure. We corrected it accordingly.
Comments: the work would be more valuable if any reproducibility data are available.
- Technical reproducibility within and between the scanners, venders
- Reproducibility in a larger group of patients
- Reproducibility between the types of myeloma
We considered your suggestion and referred to this issue in the Discussion section as a potential limitation of this approach.
Fourth, due to the small number of patients in this series and the fact that all subjects were examined on only one scanner, technical data reproducibility should be confirmed in larger patient multi-center studies.
Reviewer 3 Report
The authors describe a new way to analyze bone marrow infiltration and focal lesions in myeloma. With a new technical procedure, they correlate imaging with prognosis.
Major revision
The paper is hard to read for a clinician. It should be simplified. For example, in material and methods, part 2.2, even though these are probably important technical explanations, it would fit better in a supplemental file
Minor revisions
Ref 1 is inappropriate: it is about stem cell collection and not a general review on myeloma
Ref 2: too old, an updated one is needed
Line 56: non secretory, line 59: Salmon and Durie, line 60 : in addition to, line 69: subject to, line 86: negating?, line 93: lenalidomide
In patients characteristics, mention if patients received autotransplant or not
Line 150: twice paraproteinemia
Laboratory data: details of bone marrow infiltration are needed
Line 178: write hematological response (and not clinical, skip clinical every where)
Figure 7: x axis, PR and not PD twice
Author Response
the authors describe a new way to analyze bone marrow infiltration and focal lesions in myeloma. With a new technical procedure, they correlate imaging with prognosis.
Major revision
The paper is hard to read for a clinician. It should be simplified. For example, in material and methods, part 2.2, even though these are probably important technical explanations, it would fit better in a supplemental file
The M&M section 2.2. (MR-imaging technique) has been removed from the text body as requested and is now available as a supplementary material.
Minor revisions
Ref 1 is inappropriate: it is about stem cell collection and not a general review on myeloma
We replaced reference 1 by: Palumbo A, Anderson K. Multiple myeloma. N Engl J Med. 2011 Mar 17;364(11):1046-60.
Ref 2: too old, an updated one is needed
We replaced also reference 2 with
Bladé J, Bruno B, Mohty M. Multiple Myeloma. In: Carreras E, Dufour C, Mohty M, Kröger N, editors. The EBMT Handbook: Hematopoietic Stem Cell Transplantation and Cellular Therapies [Internet]. 7th edition. Cham (CH): Springer; 2019. Chapter 80 (80/.1).
Line 56: non secretory
Has been accordingly modified.
line 59: Salmon and Durie,
Has been accordingly interchanged.
line 60 : in addition to, line 69: subject to, line 86: negating?, line 93: lenalidomide
Corrections have been undertaken as requested.
In patients characteristics, mention if patients received autotransplant or not
Forty-nine of 67 patients underwent autologous stem cell transplantation.
Line 150: twice paraproteinemia
We meant both paraproteinemia (serum) and paraproteinuria (urin) levels.
Laboratory data: details of bone marrow infiltration are needed
We added details on the bone marrow infiltration (percentage, ranges) in the M&M section as requested.
Line 178: write hematological response (and not clinical, skip clinical every where)
We changed this term as requested throughout the whole manuscript.
Figure 7: x axis, PR and not PD twice
Sorry this was our fault. We have changed this accordingly.
Round 2
Reviewer 1 Report
The authors appropriately changed the manuscript according to the reviewers' suggestion.
Reviewer 3 Report
i have not seen the two new references (ref 1 and 2) in the revised version